# Prediction of Bonding Strength of Externally Bonded SRP Composites Using Artificial Neural Networks

**DOI:** 10.3390/ma15041314

**Published:** 2022-02-10

**Authors:** Sofija Kekez, Rafał Krzywoń

**Affiliations:** Department of Structural Engineering, Silesian University of Technology, 44100 Gliwice, Poland; sofija.kekez@polsl.pl

**Keywords:** steel-reinforced polymer, strengthening of concrete, bonding strength, artificial neural networks

## Abstract

External bonding of fiber reinforced composites is currently the most popular method of strengthening building structures. Debonding performance is critical to the effectiveness of such strengthening. Many models of bond prediction can be found in the literature. Most of them were developed based on laboratory research, therefore, their accuracy with less popular strengthening systems is limited. This manuscript presents the possibility of using a model based on neural networks to analyze and predict the debonding strength of steel-reinforced polymer (SRP) and steel-reinforced grout (SRG) composites to concrete. The model is built on the basis of laboratory testing of 328 samples obtained from the literature. The results are compared with a dozen of the most popular analytical methods for predicting the load capacity. The prediction accuracy in the neural network model is by far the best. The total correlation coefficient reaches a value of 0.913 while, for the best analytical method (Swiss standard SIA 166 model), it is 0.756. The sensitivity analysis confirmed the importance of the modulus of elasticity and the concrete strength for debonding. It is also interesting that the width of the element proved to be very important, which is probably related to the low variability of this parameter in the laboratory tests.

## 1. Introduction

High-strength fiber-reinforced composites have been used to strengthen concrete structures since the 1980s. Over the past four decades, external bonding of composites has become the most popular method of retrofitting structures, not only concrete, but also masonry, wood, and even steel. During this time, technology has improved and new types of fiber have been introduced. Fiber-reinforced polymer (FRP) composites are distinguished by an excellent strength-to-weight ratio, several times better than that of traditional steel. The currently perceived disadvantage of fiber composites is their difficult disposal and reuse. Steel-reinforced polymer (SRP) composites do not have this drawback. Although they are characterized by a slightly higher weight than carbon fiber composites, other parameters, especially tensile strength and modulus of elasticity, are comparable.

Present studies demonstrate the effectiveness of SRP composites in strengthening residential structures. After replacing the polymer matrix with cement, or even lime grout, they are perfect for repairing historic buildings. The lack of standards and design handbooks is an obstacle to the popularization of SRP composites. However, some of the procedures can be directly derived from the manuals developed for FRP composites with slightly modified mechanical properties. The issue of the bond of SRP composites may raise the greatest doubt here. SRP composites are reinforced with wire strands with a diameter of 0.25–0.35 mm. They are fixed in the wet lay-up process, which is why the fiber content in the laminate is much lower than the carbon fiber strips produced in the pultrusion process. Therefore, the relationship of the strength of SRP tapes to the cross-sectional area can even be reduced by up to three times [1], leading to a greater width and thickness of the plate. It can be expected that increasing the width would have a positive effect on the bond, but increasing the thickness would have a negative effect. The study by Mitoldis [2] showed a clear difference in adhesion and slip compared to carbon-fiber-reinforced polymer (CFRP) composites. On the other hand, Papakonstantinou [3] has shown that existing design standards can be successfully used in the design of SRP reinforced beams, but with a slightly lower safety margin. Furthermore, research [4] shows that most analytical models and design standards guarantee acceptable accuracy in predicting the bond strength of SRP composites. The results of debonding tests collected during previous analyses were used to build a model based on artificial neural networks. The created model and the results are discussed in this paper.

An artificial neural network is a machine learning technique within the wider family of artificial intelligence. It is based on the theory of connectionism, which was first proposed during the 1940s to simulate the processing of the human brain. However, the concept was not widely used until the development of information technology, which allowed its reopening and further deployment [5]. Currently, artificial neural networks (ANNs) serve for classification, i.e., the prediction of a categorical value, or regression, i.e., the prediction of a numerical value [6]. The basic concept of ANNs is grounded in the learning of patterns from the presented examples in a supervised or unsupervised manner, in other words, with or without the target values, respectively. The most used learning algorithm is the feed-forward backpropagation (BP) algorithm because of its simplicity and applicability. The BP algorithm is based on the “backpropagation learning rule” which was established in 1985 as a solution for issues occurring in single-layer or bilayer networks [6]. It represents a generalization of the delta rule and functions as a gradient descent technique of error minimization by incremental adjustment of the connection weight between the layers of a multilayer perceptron (MLP) [7]. Solving civil engineering problems conventionally involves time-consuming empirical methods or the proposal of highly complex analytical expressions. Even then, the solutions imply some type of simplification due to the limitations of the method or the complex nature of the problem. Soft computing methods such as MLPs present a time- and cost-friendly alternative to solve any problem at hand including structural health monitoring [8], structural engineering [9], gas flow [10], seismic engineering [11], etc. Furthermore, neural network-based parameter sensitivity analysis is gaining more traction in civil engineering systems due to its remarkable ability to explain the nonlinear relationships between the explicative and the response variables of a certain problem [12].

Artificial neural networks and other machine learning methods have been used to predict the behavior of elements reinforced with various FRP composite materials. Notable works include a study by Koroglu [13], which deals with the prediction of the bonding strength of FRP rebars in concrete using an ANN. The study shows the efficiency of machine learning compared to analytical methods. Furthermore, the research by Mansouri and Kisi [14] brings forward the application of ANNs and adaptive neuro-fuzzy inference systems for the prediction of debonding strength for masonry elements retrofitted with FRP composites; the works of Mashrei et al. [15], Cascardi and Micelli [16], and Jahangir and Eidgahee [17] investigated whether the application of ANNs for the prediction of bonding strength between FRP strips and concrete can give better results than the existing analytical models. It has been shown that the ANN approach may present a more efficient option. Similarly, this paper uses ANNs to predict the bonding strength of SRP composites externally bonded to concrete structures. It compares the results of ANNs with the analytical models that are traditionally used and investigates the dependencies of the ANNs variables, postulated by the learning process of the ANN, which is obtained by the sensitivity analysis of the working ANN model. The relevant information for building a comprehensive dataset was obtained from the literature [18,19,20,21,22,23,24].

## 2. Bond-Slip Models of Externally Bonded Composites

Debonding can be simply defined as the loss of bond of the composite overlay to the substrate. In practice, several ways of failure are distinguished under this phenomenon. Debonding may occur along the length of the reinforcement in the area of the highest tensile force or the anchoring zone. It can refer to the adhesive layer, but often to the contact layer in the substrate as well, or it can appear as an interlayer in the composite itself. According to Teng [25], debonding failure models can be classified as plate end interfacial debonding, concrete cover separation, intermediate flexural crack-induced debonding, and critical diagonal shear crack-induced debonding. This ordination may be complemented by debonding initiated as interlaminar failure in the matrix-fiber interface [26]. Debonding is typically simulated in a simple laboratory shear test. Depending on how the sample is clamped and how the force is applied, this may be a double or a single, push or pull-shear test. These tests directly show the bond in the end anchorage zone; however, current research proves that this testing method can also be used in the simulation of intermediate debonding [27]. An alternative and less commonly used bond test method is the notched beam test.

Bond performance is a result of the cooperation of several components with different geometries and mechanical properties. In addition, it is also influenced by the application conditions and the environmental conditions of use. Due to the number of factors and the complexity of the bond process, most mathematical bond models are based directly on the results of laboratory tests. Usually, they are classified into three categories, as follows:fully empirical models, based on the regression of test data, such as Tanaka [28], Hiroyuki and Wu [29], and Maeda [30],fracture mechanics-based models including Taljsten [31], Niedermeier [32], Yuan and Wu [33], and Lu et al. [34],design models, usually based on simple assumptions, such as Dai et al. [35], Brosens and van Germet [36], Khalifa et al. [37], Yang et al. [38], Adhikary and Mutsuyoshi [39], Sato et al. [40], Chen and Teng [41], DeLorenzis et al. [42], and Seracino et al. [43].

A detailed description of these models can be found inter alia in Table 1 [4].

Evaluation of the above-mentioned bond models for SRP strengthening (carried out by this author) showed that most of them are in relatively good agreement with the test results [4]. As can be seen in Table 1, the best prediction accuracy is given by Lu et al. [34], Chen and Teng [41], and most of the design standards.

When analyzing the predicted experimental ratio, most models can be considered rather conservative. Some models significantly increase the load capacity (including Dai et al. [35], Brosens and Germet [36], and Adhikary and Mitsuyoshi [39]). The largest scatter in the results concerns the proposals by Dai et al. [35], Adhikary and Mitsuyoshi [39], and DeLorenzis et al. [42]. The common feature of these models is the independence of the effective anchorage length.

## 3. Artificial Neural Networks

Artificial neural network (ANN) models have been developed to predict the bonding strength of the externally bonded SRP composite to the concrete element. All models are feed-forward backpropagation networks with a sigmoid activation function and a linear transfer function. Data were collected from the works of Figeys et al. [18], Matana et al. [19], Mitoldis et al. [20], Napoli et al. [21], and Ascione et al. [22,23,24]. Table 2 briefly summarizes the most relevant information from these studies, which are also incorporated in the dataset. The comprehensive dataset was halved, so that the first half could be used for training and the other half is used for testing the ANN model. The dividing of the dataset represented the realistic behavior of the established model, i.e., the level of fitting with the unseen data by the network. Finally, the working model was established and trained using the entire dataset.

The entire dataset was randomly divided while ensuring that each half included data from every referenced source. Both the training and the testing sets consisted of 171 data tuples. Input data included information on the following: sample width, sample thickness, concrete compressive strength, concrete tensile strength, concrete modulus of elasticity, tape width, tape thickness, anchorage length of the tape, modulus of elasticity of SRP, the tensile strength of SRP, and the type of adhesive between the concrete element and the SRP tape. The target was the debonding force of the SRP tape from the concrete element. All input data, except the type of adhesive, were numerical. The numerical values of the input and target data were processed with min/max normalization within the [0, 1] range and, as such, were presented to the network. The types of adhesives between the SRP tape and the concrete element were epoxy and grout, presented to the network as the values 0 and 1, respectively. Figure 1 shows the distribution of values of the numerical input parameters throughout the entire dataset.

### 3.1. Establishing the Working ANN Model

Neural network models were developed using Matlab R2020b. The supervised training of the models was carried out with the following learning parameters: unipolar sigmoid activation function, linear transfer function, Levenberg–Marquardt algorithm, one hidden layer, 1000 epoch limit, 10^9^ momentum, 10^−6^ learning rate, and six-fold cross-validation. Training was set to stop when the network did not improve after six consecutive validation checks. The procedure for establishing a working ANN model for the prediction of bonding strength was set as follows. Firstly, the training was performed on the initial model. The optimization of the initial model was then provided and the training repeated for the optimized model. The testing was performed on the optimized model to establish the quality of generalization. Since the optimized model was tested on new unseen data, it may be assumed that the results would show the realistic behavior of the network. Finally, the final working ANN model was established and trained on the full dataset.

#### 3.1.1. Initial Model

The initial model included the simplest network architecture, having the input layer with eleven neurons, one hidden layer with an equal number of neurons as the input layer, and the output layer with one neuron. The training set was pre-processed using min/max normalization with the [0, 1] range for the numerical values, and zero or one values for the descriptive input. During the training process, 15% of the set was used for validation to ensure that the training process stopped when the six consecutive validation checkpoints showed no improvement in the training.

#### 3.1.2. Optimization of the Initial Model

Optimization of the initial model served to establish the optimal number of neurons in the hidden layer. An improved topology of the network implied a better generalization and contributed to the stability of the network. Optimization was indicated by the level and change of the mean squared error for the varying number of neurons in the hidden layer. To this end, consecutive training was performed for models with an iterated number of neurons from one to fifty in the hidden layer.

The literature gives general recommendations on the number of neurons in the hidden layer. It is considered that for the network of this size in terms of the input neurons and the batch size, a very high number of neurons in the hidden layer would surely cause an overfitting of the network. The overfitting leaves the network unable to generalize and perform the prediction when presented with the new data. Hence, it is considered that the highest number of neurons in the hidden layer should not exceed fifty. After the iterations finished, the change in the mean squared error (MSE) was observed for all iterations and compared to establish which number of neurons in the hidden layer gave the satisfactory results and exhibited stability. On the other hand, it is important to secure the network from the occurrence of underfitting as well. Practically, the lowest number of neurons in the hidden layer may not be less than the number of the input neurons. Furthermore, the literature often gave the recommendation that the number of neurons in the hidden layer should not be under Ni+2, where Ni is the number of input neurons. Underfitting may be spotted as the occurrence of extremely low MSE, giving an overly positive result of the network’s behavior.

#### 3.1.3. Training and Testing of the Optimized Model

When the optimization process was concluded and the optimal network topology set, the new model was then independently trained with the previously described training set. The training was performed with the same learning parameters as the initial ANN model. The testing of the optimized model followed the training process. It was performed using the second half of the dataset, which presents entirely new data for the neural network. Hence, the testing process can show the real behavior of the network, i.e., the capability of generalization when the network is presented with unknown data. It is expected that proper generalization implies good prediction and similar results, after training and testing, may confirm the stability of the network.

#### 3.1.4. Working ANN Model

The final ANN model presents the working model which can be further used for predictions of the bonding strength of the externally bonded SRP composite to the concrete element. This model used the previously established and tested topology of the optimized ANN model; however, it was trained using the entire dataset. After the successful training process, the weights and bias were fixed, and the network presented a ready-to-use model.

### 3.2. Sensitivity Analysis

The sensitivity analysis served to show the absolute or relative contribution of each input parameter to the output value. It was necessary to understand the relationship and influence of the input parameters on the problem that the ANN learns to solve [7]. Except for showing the contribution of each input parameter, the sensitivity analysis may also influence the topology of the final working model, because it may show that some parameters could hinder or slow down the learning process. On the other hand, it shows which parameters are crucial for the learning process, as well as the dependence of the output value on each input parameter. This analysis was provided using the weights method, otherwise known as the Garson’s algorithm [48]. The algorithm was created for supervised neural networks with a single output, to describe the relative importance of the input parameters by deconstructing the model weights. The mathematical description of the algorithm for a network with a single hidden layer is as follows:(1)$$D_{ij}=\frac{\left|W_{ij}\right|}{\sum_{i=1}^{n_{i}}\left|W_{ij}\right|}$$
(2)$$RC_{i}=\frac{\sum_{j=1}^{n_{j}}D_{ij}}{\sum_{j=1}^{n_{j}}\sum_{i=1}^{n_{i}}D_{ij}}$$
where *n_i_* and *n_j_* are the numbers of input and hidden neurons, respectively, *W_ij_* is the weight corresponding to the *i*-th input and the *j*-th hidden neuron, and *RC_i_* is the relative importance of the *i*-th input.

## 4. Results

The performance of the ANN model is described by the mean squared error MSE, root mean squared error RMSE, and the regression coefficient R. The MSE and RMSE represent the average squared and average root squared difference between the output and the target value, respectively, which tends to zero as the prediction becomes more accurate. The R value is the primary parameter that shows the correlation of the output compared to the target value, which tends to a value of one, as the prediction becomes more accurate. The regression coefficient is usually expressed as the total of R values for training, testing, and validation. Additionally, the error distribution shows the general behavior of the network and the relationship between the error during the training, testing, and validation.

### 4.1. Training and Optimization of the Initial ANN Model

As mentioned previously, the initial model was trained with the architecture including eleven neurons in the input and the hidden layer and a single neuron in the output layer. The results of the initial training show that there is room for improvement. Table 3 shows that, although the RMSE value is quite low, the regression coefficient of 0.85 for training implies that the learning process should be improved. Figure 2 shows the relationship between the target and the output values after training the initial model.

Optimization of the initial model has been performed by simply iterating the number of neurons in the hidden layer and observing the MSE for training and validation. To establish the optimal number of neurons in the hidden layer, fifty iterations were carried out. The results of the optimization process are presented in Figure 3. It shows that overfitting occurs with 49 neurons in the hidden layer. Vis-à-vis, underfitting may have occurred with 5, 8, 9, 12, and 13 neurons in the hidden layer. The literature recommends the highest number to be equal to 2*Ni +* 1, where *Ni* is the number of input neurons. Thus, the preferred number of hidden neurons should be between fifteen and thirty, which is supported by the results of the optimization of the network. A sudden drop or increase in the error for consecutive iterations may imply instability; hence, the error should show a relatively minimal change for several consecutive iterations. The behavior is somewhat steady within the range of 18 to 23 neurons, especially when the training MSE values are observed. Furthermore, within this range, the closest result between the training and the validation MSE occurs when the number of hidden neurons is 21, and thus it is considered to be an optimal value.

### 4.2. Training the Optimized ANN Model

The optimized ANN model has been trained in exactly the same manner as the initial model, so that a realistic comparison between the two models can be achieved. The only difference is the number of neurons in the hidden layer, which is equal to 21 for the optimized model. Table 3 shows the comparison of the regression coefficients and the mean squared error between the initial and optimized models. The improvement is visible; the error of the optimized model shows a value closer to zero and regression coefficients show an improvement in the learning process and the prediction accuracy. Figure 4 compares the error distributions for both models. It may be observed that the error distribution of the optimized model exhibits a more uniform decrease in the error with less outliers and more symmetrical distribution around zero error.

Figure 5 compares the relationship between the target and the output values for the initial and the optimized model after training. The improvement is reflected in the significant increase in the training regression coefficient after optimization. Moreover, the validation R value shows a much higher increase which, in turn, gives the overall R value of the optimized model equal to almost 0.87. This implies a better prediction accuracy of the optimized model in comparison to the initial prediction. It may be assumed that the final model will show even better results because the training will be provided with twice as many data tuples.

### 4.3. Testing the Optimized ANN Model

After it had been concluded that the optimization of the initial model was successful, testing by introducing the data which the network had never seen presented the final verification for the optimized ANN model. This consists of introducing the second half of the dataset, data that were randomly selected during the division of the set, to the network. The testing of the trained network was performed by introducing the new set and calling a simulation function to the trained network. The results are presented by relating the target and the output values, as shown in Figure 6. The total R value after testing exceeds 0.91, which implies great success in the network generalization capability. Figure 7 shows the comparison of the results after training and after testing the optimized network. The network exhibits very similar behavior after training and testing, indicating that the architecture is suitable and that the network is stable with good generalization. The prediction accuracy is at a satisfactory level, given that the data were obtained through different sources, and the input data were somewhat repetitive.

### 4.4. Training of the Working ANN Model

The final ANN model was established, tested, and confirmed to be successful in predicting the bond strength of the externally bonded SRP composite to the concrete element. In order to obtain the working model, the neural network was trained once more, using the entire dataset. Then, the weights and bias were set and fixed, and the neural network used for prediction. The learning parameters were kept from the previous models, the number of neurons in the hidden layer was 21, and the subset ratio was 80/15/5 for training/testing/validation. Figure 8 shows the error distribution and the relationship between the target and the output values, while Table 4 shows the values of the regression coefficients and the mean squared error for the working ANN model. The improvement is best seen when observing the error distribution which takes a Gaussian zero-centered shape. The lack of outliers is visible in both Figure 8a,b and, lastly, the regression coefficients experience a significant increase. It may be observed that the prediction accuracy of the working model exceeds 90%, which indicates a very reliable neural network.

### 4.5. Sensitivity Analysis

The sensitivity analysis of the working ANN model has been provided using Garson’s algorithm, i.e., the weights method, according to Equations (1) and (2). Table 5 shows the values of the weights connecting the neurons in the input and the hidden layer. These values are used to calculate and determine the relative contribution of each input parameter regarding the output. Figure 9 and Figure 10 show the contribution of the input parameters. The results of the analysis show that all input parameters have a very close level of importance to the output. Relatively speaking, the modulus of elasticity of the SRP composite and the width of the concrete element have the highest importance to the output. Analysis shows that the anchorage length of the SRP composite is the least important to the bonding strength. Additionally, the thickness of the concrete element and the compressive strength of the concrete show low importance regarding the bonding strength for externally bonded SRP composites to the concrete element. However, none of the input parameters show less than a 50% contribution to the output value, meaning that none of them may be excluded from the dataset.

## 5. Discussion of Results

The comparison of Table 1 and Table 4 clearly shows that the ANN method guarantees a much better quality of results than any of the analytical methods evaluated. The total correlation coefficient R_total_ was equal to 0.91338, which is more than satisfactory and only slightly lower than the one obtained by the authors of a similar analysis of the adhesion of CFRP composites [15,17]. However, it should be noted that, in the case of CFRP composites, due to their popularity, much larger datasets of research results are disposable, which directly affects the behavior of the ANN model.

Mukhopadhyaya and Swamy [49] pointed out that increasing the elastic modulus of the bonded fiber composite results in a higher value of interfacial stresses. Teng [25], who also noted that the elastic modulus does not affect the location of the peak value, found similar conclusions. The authors cited also indicate similar findings regarding the thickness of the composite. Similarly, an additional layer of laminates increase stress (Shahawy et al. [50]). It is proof of the key role of laminate stiffness in interfacial stress values and, thus, the probability of premature debonding. The described effect has been proven by the sensitivity analysis in this work. The modulus of elasticity *E_f_* and the thickness of the laminate *t_f_* are the most important input parameters (Figure 10).

In most cases, delamination occurs in the contact layer between the adhesive and the concrete or the cover layer; therefore, it is commonly considered that the bond is significantly affected by the strength of the concrete and the preparation of the concrete surface [51]. The concrete tensile strength *f_ct_* is the fourth most important parameter indicated by the sensitivity analysis. A similar meaning can be assigned to the *Ad* parameter, which represents the type of adhesive. Admittedly, the mechanical properties of the adhesive were not entered into the model. Only the division into epoxy resin and grout was parameterized. Epoxy adhesives are characterized by much better strength and bond properties, which were rightly indicated in the analysis.

Surprisingly, the lowest impact of the bond length *L* may be caused by the specificity of the test data. Most analytical methods define the effective bond length along which most of the interfacial load is transferred. For the bond length, which exceeds the *L_e_*, the bond strength does not increase significantly. The effective bond length depends mainly on the stiffness of the composite plate. There is no consensus as to what this length is; for example, Sato [40] gives values of around 45 mm, while Brosens and Germet [36] suggest over 275 mm. The effective bond length for a single layer of SRP, calculated according to the above-mentioned analytical methods [30,32,41,45,47], is in the range of 50–160 mm and for most cases does not exceed the anchorage length provided in the test samples (Table 2 and Figure 1). Therefore, the slight influence of the anchorage length observed confirms a limited transfer of the interfacial force over the effective bond length.

The second result that requires comment is the significant effect of the sample width *b*. This problem may be related to the ratio of the composite width *b_f_* to the sample width *b*, as this influences the stress distribution in the concrete. For the tested models, this proportion is usually equal to 0.5; therefore, slight changes in the width *b* could significantly affect the distribution of stresses and, thus, the debonding strength.

## 6. Conclusions

This paper describes an innovative approach for estimating the bond strength of SRP to concrete, based on the artificial neural network model. The developed model is trained on the basis of the experimental data gathered from published literature. The model is used to predict the bonding strength and further compared to some of the analytical bond-slip models from the literature. The results obtained show good agreement with the laboratory data collected. The working ANN model performs significantly better than other models in estimating the bonding strength. The sensitivity analysis concludes that the architecture of the working model is also optimal in terms of the number of input neurons. None of the input parameters can be excluded from the network, as all of them carry a high level of importance to the output value.

Undoubtedly, the study shows the potential of neural networks as a supporting tool for structural engineers; however, the main disadvantage of this method is that it is a ‘black box’ that cannot derive any universal equation and cannot function without a training base. On the other hand, it is only a matter of time before such datasets will be automatically created by the internet bots.

## Figures and Tables

**Figure 1 materials-15-01314-f001:**
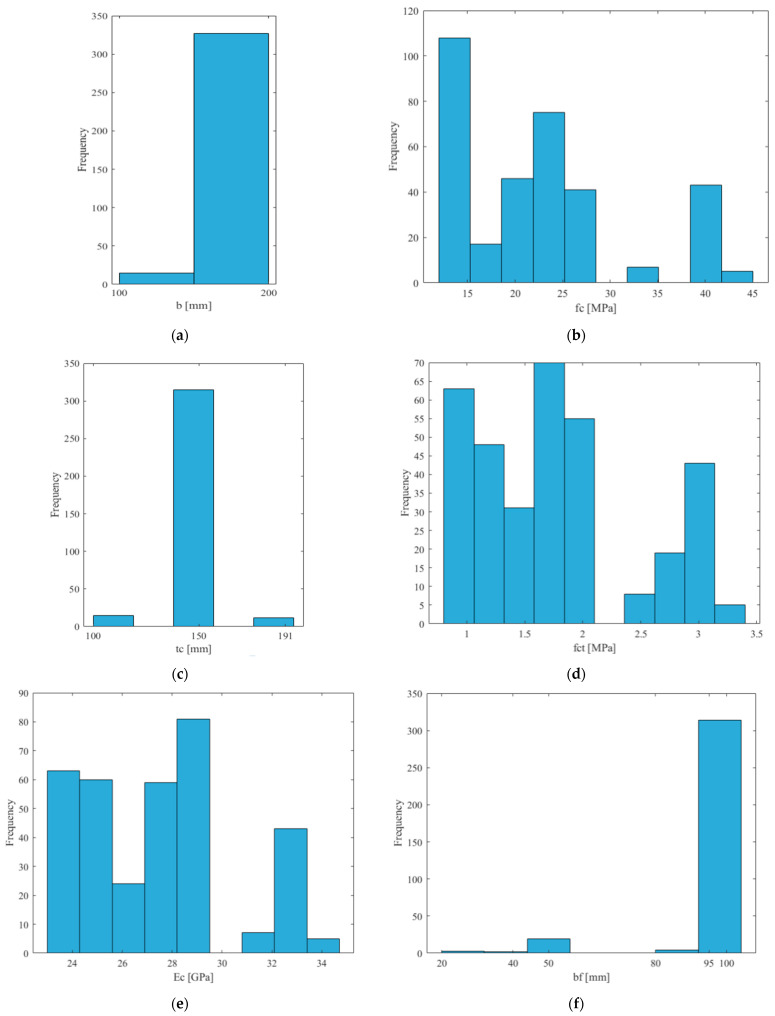
Input parameters: (**a**) sample width—*b*; (**b**) concrete compressive strength—*f_c_*; (**c**) sample thickness—*t_c_*; (**d**) concrete tensile strength—*f_ct_*; (**e**) concrete modulus of elasticity—*E_c_*; (**f**) SRP tape width—*b_f_*; (**g**) anchorage length—*L*; (**h**) SRP tape thickness—*t_f_*; (**i**) SRP modulus of elasticity—*E_f_*; (**j**) SRP strength—*f_f_*.

**Figure 2 materials-15-01314-f002:**
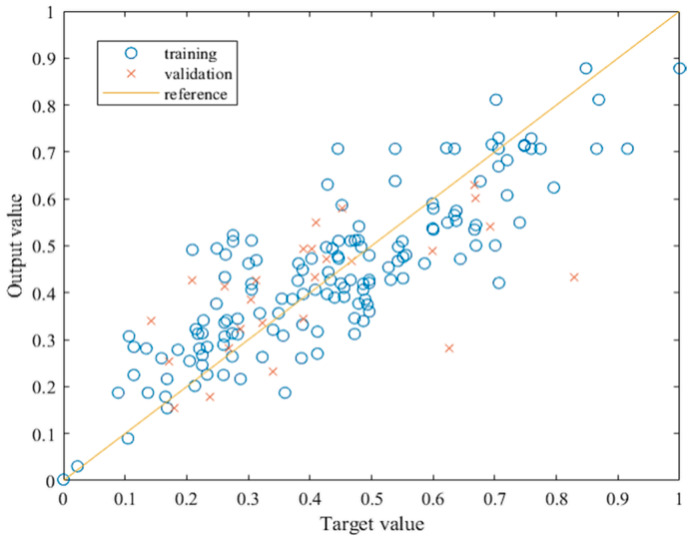
Relationship between the target and output values of the initial model.

**Figure 3 materials-15-01314-f003:**
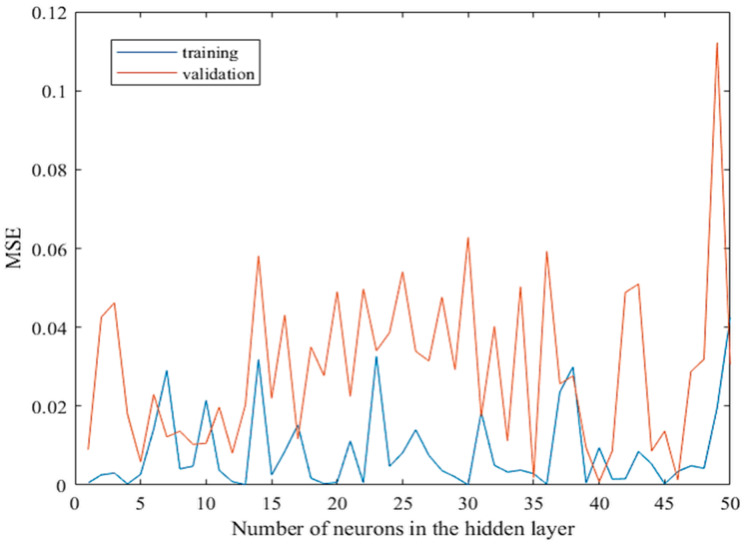
MSE values after optimization of the initial model.

**Figure 4 materials-15-01314-f004:**
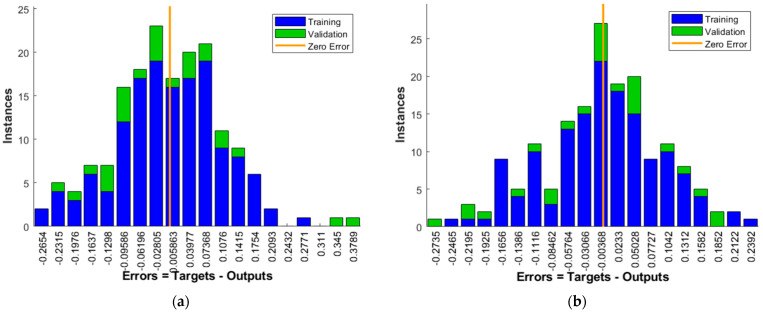
Error histogram for: (**a**) Initial model; (**b**) Optimized model.

**Figure 5 materials-15-01314-f005:**
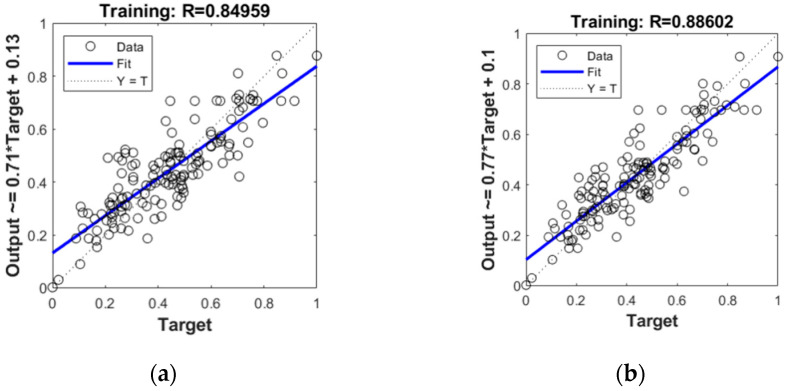
Regression after training for: (**a**) Initial model; (**b**) Optimized model.

**Figure 6 materials-15-01314-f006:**
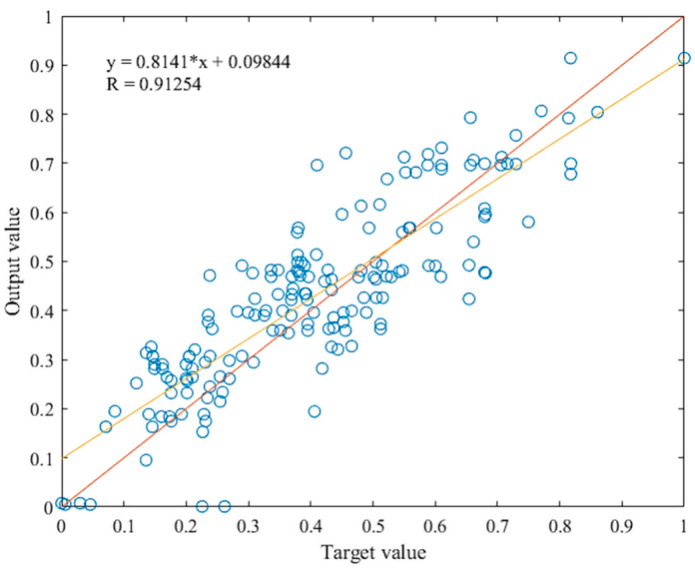
Relationship between target and output value after testing.

**Figure 7 materials-15-01314-f007:**
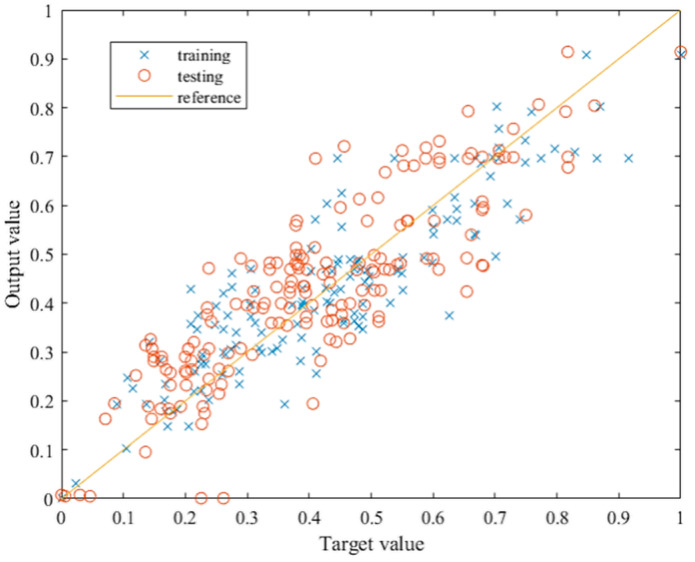
Relationship between target and output values after training and testing.

**Figure 8 materials-15-01314-f008:**
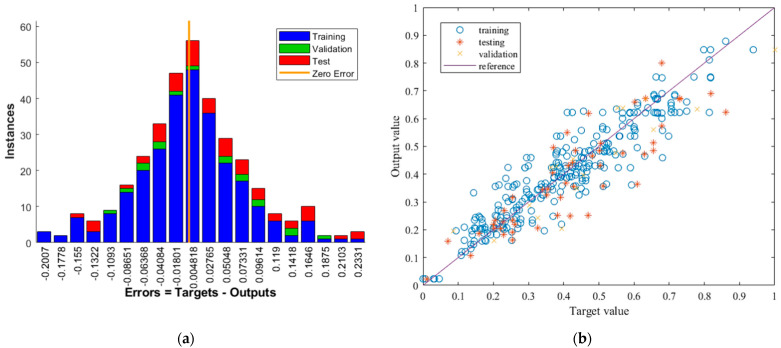
(**a**) Error distribution for the working model; (**b**) Relationship between target and output values for the working model.

**Figure 9 materials-15-01314-f009:**
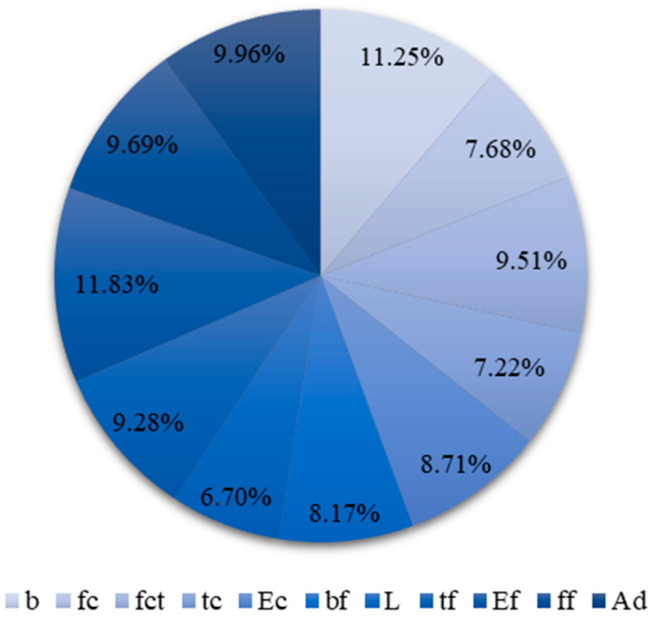
Contribution of the input parameters.

**Figure 10 materials-15-01314-f010:**
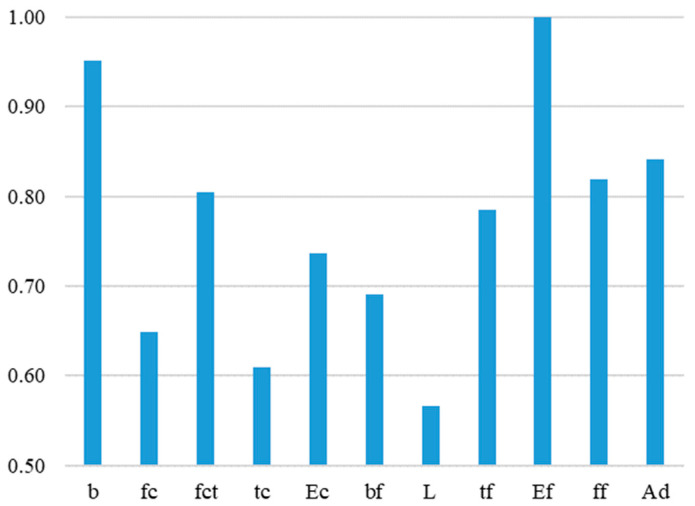
Relative importance of the input parameters.

**Table 1 materials-15-01314-t001:** Predicted experimental bond strength ratios for SRP composites bonded to concrete [4].

Model	Mean	R
Tanaka [28]	0.699	0.220
Hiroyuki and Wu [29]	0.746	0.503
Maeda et al. [30]	1.14	0.751
Taljsten [31]	0.761	0.748
Nidermeier [32]	0.763	0.629
Yuan and Wu [33]	0.763	0.747
Lu et al. [34]	0.873	0.676
Dai et al. [35]	1.55	0.734
Brosens and van Germet [36]	0.937	0.408
Khalifa et al. [37]	0.754	0.729
Yang et al. [38]	0.528	0.657
Adhikary and Mutsuyoshi [39]	2.00	0.476
Sato et al. [40]	1.81	0.573
Chen and Teng [41]	0.882	0.726
DeLorenzis et al. [42]	1.63	0.728
Seracino et al. [43]	0.752	0.716
JCI 2003 [44]	0.941	0.738
SIA 166/2004 [45]	0.911	0.756
CNR-DT200R1/13 [46]	0.928	0.718
Fib Bulletin 90/2019 [47]	0.962	0.755

**Table 2 materials-15-01314-t002:** List of geometry and material properties collected from the available experimental data.

Reference	Number of Specimens	*b* [mm]	*f_c_* [MPa]	*b_f_* [mm]	*t_f_* [mm]	*L* [mm]	*E_f_* [GPa]
Figeys [18] ^1^	7	100	35	95	0.601	150–200	177.6
Mantana [19] ^1^	12	191	14.8	51	0.483	102–305	179.1
Mitoldis [20] ^1^	8	100	22.4	50–80	0.562	150–300	221.4
Napoli [21] ^1^	19	200	15.2–39.7	100	0.084–0.381	150–300	206.6
Ascione [22] ^1^	129	200	13–45	20–100	0.084–0.381	100–350	190
Ascione [23] ^1^	62	200	19.3–25.6	100	0.084–0.381	100–350	182.1–183.4
Ascione [24] ^2^	83	200	13–40	50–100	0.084–0.254	100–350	182.1–183.4

^1^ epoxy adhesive; ^2^ grout adhesive; *b*—sample width; *b_f_*—width of SRP tape; *f_c_*—concrete strength; *t_f_*—effective SRP thickness; *L*—bond length; *E_f_*—modulus of elasticity of SRP.

**Table 3 materials-15-01314-t003:** Comparison between the initial and the optimized ANN model.

Model	R Training	R Validation	R Total	MSE	RMSE
Initial	0.84959	0.60249	0.82339	0.0106	0.00991
Optimized	0.88602	0.75031	0.8675	0.0099	0.0076

**Table 4 materials-15-01314-t004:** Results of the working ANN model.

Model	R Training	R Validation	R Testing	R Total	MSE	RMSE
Working	0.92367	0.90783	0.87864	0.91338	0.0073	0.00023

**Table 5 materials-15-01314-t005:** Connection weights between the neurons in the input and the hidden layer.

Hidden/Input	1	2	3	4	5	6	7	8	9	10	11
1	−0.636	0.707	−0.534	−0.392	0.487	−0.315	−0.667	−0.350	0.735	−0.294	0.151
2	0.220	−0.179	−0.942	0.458	−0.140	0.648	−0.123	0.243	1.164	0.768	−0.725
3	0.869	−0.322	0.604	0.374	−0.508	−0.695	0.225	−0.248	0.683	0.792	0.612
4	0.808	−0.866	−0.555	−0.363	−0.298	−0.214	0.564	0.361	−0.586	0.244	−0.321
5	0.937	0.329	0.305	0.125	0.641	0.328	−0.399	−1.250	0.453	0.501	−0.474
6	−0.612	−0.658	0.017	0.398	0.309	−0.303	−0.459	−0.706	−0.617	0.799	0.622
7	0.026	−0.190	−0.592	0.073	−0.202	−0.638	−0.662	−0.539	−0.247	−0.402	−0.796
8	−0.745	0.763	0.816	0.029	−0.570	−0.670	0.096	0.240	−0.762	−0.548	0.257
9	0.374	0.309	0.272	−0.685	0.694	−0.344	−0.631	0.571	0.739	−0.310	0.234
10	−0.789	−0.136	0.101	0.353	−0.799	0.067	−0.632	−0.826	−0.024	0.026	−0.464
11	0.297	−0.317	−0.279	0.447	−0.384	0.813	−0.200	−0.520	0.846	−0.641	0.443
12	−0.195	−0.258	−0.439	0.809	0.615	0.047	0.045	−0.139	1.389	−1.057	1.407
13	−0.969	−0.416	0.948	−0.283	0.186	−0.755	0.097	−0.434	0.055	−0.269	−0.401
14	0.738	0.347	0.001	−0.245	−0.101	0.154	0.046	−0.110	−0.466	1.051	0.266
15	0.584	−0.654	−0.688	−0.625	0.424	0.229	0.231	0.091	0.573	0.209	−0.645
16	−0.701	0.048	0.123	0.455	−0.953	0.754	−0.150	0.071	−0.883	0.602	−0.331
17	0.909	−0.300	−0.557	−0.173	−0.024	−0.199	1.039	0.753	0.426	−0.065	0.451
18	−0.500	−0.339	−1.039	−0.644	−0.327	0.003	−0.076	−1.405	0.507	−0.472	0.472
19	0.321	−0.838	−0.861	0.110	0.399	0.605	−0.564	0.524	0.567	−0.955	0.532
20	0.206	0.258	−0.391	−0.450	0.749	−0.818	0.283	0.408	0.222	0.225	−0.370
21	0.901	0.192	0.369	−0.427	0.743	0.360	0.161	0.392	−1.031	−0.399	−0.945

## Data Availability

The data presented in this study are available on request from the corresponding author.

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
