# Peer review of "Prediction of Bonding Strength of Externally Bonded SRP Composites Using Artificial Neural Networks"

_materials, 2022, doi:10.3390/ma15041314_

Round 1

Reviewer 1 Report

The manuscript entitled

”Prediction of Bonding Strength of Externally Bonded SRP Composites Using Artificial Neural Networks “

has been investigated in detail. The topic addressed in the manuscript is potentially interesting and the manuscript contains some practical meanings. The literature of the manuscript should be modified. Also, there are some issues which should be addressed by the authors:

  • Fig 3. Shows the MSE versus the number of neurons in the hidden layer. The authors know that every MLP network reaches to specific result in every specific run. In Fact, one network with a specific architecture has 2 different results in 2 different runs! So I think this figure is not useful for readers.

  • “Roshani, G.H., et al., 2017. Flow regime independent volume fraction estimation in three-phase flows using dual-energy broad beam technique and artificial neural network. Neural Computing and Applications, 28(1), pp.1265-1274” and “Sattari, M.A.et al. Applicability of time-domain feature extraction methods and artificial intelligence in two-phase flow meters based on gamma-ray absorption technique. Measurement 2021, 168, 108474, doi:10.1016/j.measurement.2020.108474.” could be used in the study especially regarding the proposed neural network based model.

  • How was the best architecture (number of hidden layers, epochs and etc.) of network obtained?

  • Why do the authors use only MLP network? The authors could compare the mentioned method with other efficient and updated methods.

  • Other different errors such as MRE, MAE and RMSE could be used.

  • Why the overfitting problem has not been occurred?

This study may be consider for publication if it is addressed in the specified problems.

Author Response

Dear reviewer. Thanks for your comments. We have done our best to implement the suggested manuscript corrections. The answers to the questions and concerns raised in your review are below.

  1. Fig 3. Shows the MSE versus the number of neurons in the hidden layer. The authors know that every MLP network reaches to specific result in every specific run. In Fact, one network with a specific architecture has 2 different results in 2 different runs! So I think this figure is not useful for readers.

Yes, of course, the results differ in each run. However, the goal here is to show the behavior of the network relatively, in terms of the number of hidden neurons as well as the validation and training errors. For that reason, the concrete values of MSE where not discussed within the process of obtaining the final number of hidden neurons. Figure 3 shows us the occurrence of overfitting and underfitting, and which range of hidden neurons implies most stability. This is additionally addressed within the section 3.1.2. and 4.1. of the paper.

  1. Roshani, G.H., et al., 2017. Flow regime independent volume fraction estimation in three-phase flows using dual-energy broad beam technique and artificial neural network. Neural Computing and Applications, 28(1), pp.1265-1274” and “Sattari, M.A.et al. Applicability of time-domain feature extraction methods and artificial intelligence in two-phase flow meters based on gamma-ray absorption technique. Measurement 2021, 168, 108474, doi:10.1016/j.measurement.2020.108474.” could be used in the study especially regarding the proposed neural network based model.

Thank you for your suggestion, the proposed literature has been reviewed.

  1. How was the best architecture (number of hidden layers, epochs and etc.) of network obtained? Why do the authors use only MLP network? The authors could compare the mentioned method with other efficient and updated methods.

The architecture is assumed as the simplest shallow network with the default prerequisites to observe the possibilities for the prediction as the pointer for further improvements. Additionally, the applicability of MLPs is a strong advantage for practical use in civil engineering.

  1. Other different errors such as MRE, MAE and RMSE could be used.

RMSE from the same network run has been added to the section 4.

  1. Why the overfitting problem has not been occurred?

Overfitting has occurred with the number of neurons in the hidden layer equal to 49, as it is discussed within section 4.1.

Reviewer 2 Report

This manuscript applied an ANN model to predict the bonding strength between EB SPR and concrete based on the obtained experimental data. The establishing process of the ANN models is very careful, including an initial model, an optimized model, and a working model. Overall, the manuscript is well organized. I just have some questions:

  1. Some analytical models in Table 1 were aimed at the bonding strength of EB or NSM FRP to concrete joints (e.g., Ref[30, 33, 34, 35, 39]). Can they be directly used to predict the strength between SPR and concrete? Their prediction accuracy can not be high since they are established for other materials. The authors should explain this.
  2. The choice of the number of the hidden layer is worth further thinking. As shown in Figure 3, 21 neurons in the hidden layer do not outperform other cases. For example, 12 neurons show a lower MSE both in the training and validation process.

Author Response

Dear reviewer. Thanks for your comments. We have done our best to implement the suggested manuscript corrections. The answers to the questions and concerns raised in your review are below.

  1. Some analytical models in Table 1 were aimed at the bonding strength of EB or NSM FRP to concrete joints (e.g., Ref[30, 33, 34, 35, 39]). Can they be directly used to predict the strength between SPR and concrete? Their prediction accuracy can not be high since they are established for other materials. The authors should explain this.

A very valid point. This problem was discussed in more detail in earlier studies (cited in the manuscript as [4]). Most of the methods of determining the bond were developed almost 20 years ago and mainly concerned composites reinforced with carbon fibers, less glass ones. SRP composites were developed a bit later, however, from a technical point of view, they are not much different from CFRP composites (similar modulus of elasticity, similar strength). It can even be said that they are between the CFRP sheet and the CFRP strip. Larger differences concern glass or aramid composites, which are also calculated according to the same principles. The low quality of prediction shown in Table 1 concerns mainly older methods, developed on the basis of a limited number of samples. We have currently collected a database of over 800 CFRP bond tests and the results are even worse than in the case of SRP (for example, the average prediction for Tanaka is 0.47, Hiroyuki 0.65)

  1. The choice of the number of the hidden layer is worth further thinking. As shown in Figure 3, 21 neurons in the hidden layer do not outperform other cases. For example, 12 neurons show a lower MSE both in the training and validation process.

Thank you for your observation. However, in order to avoid the problem of underfitting, we have observed which interval showed the most stable behavior, and where the validation and training MSE showed similar values. It could be considered that for the Nh=12, underfitting is occurring due to extremely low MSE while the number of hidden and input neurons is almost equal. The comment is addressed in the section 3.1.2. and further in the section 4.1.

Following your suggestions, the manuscript has been revised in English by a professional proofreading agency.

Round 2

Reviewer 1 Report

All of the comments and corrections have been applied in the manuscript. My recommendation is accept.